# Adaptive Ultrasonic Full Matrix Capture Process for the Global Imaging of Complex Components with Curved Surfaces

**DOI:** 10.3390/s24010225

**Published:** 2023-12-30

**Authors:** Wensong Miao, Ne Liu, Jingqiang Huang, Minghui Lu

**Affiliations:** 1Key Laboratory of Nondestructive Testing of the Ministry of Education, Nanchang Hangkong University, Nanchang 330063, China; 2108085400022@stu.nchu.edu.cn (W.M.); 2208085400003@stu.nchu.edu.cn (J.H.); 2Faculty of engineering, University of Sydney, J12/1 Cleveland St, Darlington, NSW 2008, Australia; nliu4612@uni.sydney.edu.au

**Keywords:** ultrasonic testing, full matrix imaging, surface adaptation

## Abstract

This work proposes a new global FD-RTM method to solve the problem of ultrasonic inspection of parts with complex geometric shapes. With this method, the frequency domain reverse time migration (FD-RTM) algorithm is used to adapt to the complex refraction of ultrasonic waves by the surface, while an interface solution algorithm based on tangent fitting is used to solve the interface position with high precision through the full matrix reception data. Based on high-precision interface information, a hybrid extrapolation algorithm and a situation-specific probe movement strategy are used to enable the probe to find the next sampling point according to the direction of the workpiece surface, allowing complex surface topography features to be identified without relying on the workpiece CAD drawing. This makes it possible to achieve the automated inspection of workpieces. To verify the proposed method’s effectiveness, an aluminum alloy model with side-drilled holes (SDH) is used. The geometry of the model consists of multiple convex and concave surfaces. By comparing the local FD-RTM imaging with images synthesized using the entire scan path, it is shown that gFD-RTM improved the imaging performance. Compared with FD-RTM, the average signal-to-noise ratio of gFD-RTM was increased by 20%, and the array performance index (API) was reduced by 70%, indicating effective detection coverage.

## 1. Introduction

Curved surface components are widely used in various fields. For example, shafts and cams are the most important components of drive systems [1]. Due to the aerodynamic requirements, curved components in the aerospace [2], automotive [3,4] and energy [5] fields are highly complex. Because of the complex manufacturing processes and harsh use environments, curved components are prone to voids and crack defects, which greatly affect the mechanical properties of the components, leading to premature component failure and serious accidents [6,7]. Therefore, the detection of internal defects in curved components is a problem that urgently needs to be solved. However, large-scale curved components and complex curved surfaces pose immense challenges to detection technology.

Ultrasound imaging methods using phased arrays have gained prominence in non-destructive testing and have proven their effectiveness in characterizing invisible defects. Each element in a phased array acts as a transceiver, and full matrix capture (FMC) collects the complete signal from each pair of transmit and receive elements [8]. Previous researchers have used flexible probes [9], ice coupling [10] and other methods to conduct FMC-based ultrasonic testing of curved surfaces. The most economical and convenient method is to immerse the workpiece and a flat probe in water and use an imaging algorithm adapted to the curved surface for imaging [11,12,13]. Among the existing FMC imaging methods, frequency domain reverse time migration (FD-RTM) is a new technology with the smallest artifacts and the highest resolution [14], and it is the imaging technology used in this article.

FMC imaging methods, including FD-RTM, are limited to imaging with fixed-position probes. When inspecting large workpieces, it is necessary to move the probe for sampling and imaging at multiple positions to obtain complete non-destructive inspection results for the workpiece. For large workpieces with curved surfaces, the path planning of the automation system is particularly important [15]. At present, the main methods to obtain accurate surface component drawings can be divided into manual [16], direct import of CAD drawings [15,17] or reverse engineering of curved surface components through professional measuring equipment such as three-axis measuring instruments, lidar or structure light [18,19,20,21]. However, the CAD drawing of a workpiece may not be available, and the use of external equipment will increase costs. Therefore, the best method is to use a probe for flaw detection to directly measure the surface. This method is divided into manual operation and an automated system marking sampling points on the workpiece [22] and probe movement strategy to allow the inspection system to measure the entire workpiece by itself [23,24,25,26]. An automated probe will consume less time and labor costs. Currently, both the manual and automated probe methods involve the movement of a single probe to obtain a series of two-dimensional data points. In contrast, the imaging algorithm used in this article requires FMC data to be collected through an array probe so that more interface information can be calculated from the FMC data.

This study combines an automated system and phased array equipment with a new post-processing FMC imaging method called global FD-RTM (gFD-RTM). The proposed gFD-RTM method allows the imaging of the entire cross-section of complex samples and automatically adapts to surface contours without relying on CAD drawings, thereby extending the capabilities of FD-RTM. This is possible because gFD-RTM combines water-immersed phased array ultrasonic inspection with automated equipment and therefore allows complex and sophisticated scan plans. Moreover, the proposed approach enables the adaptive image scanning of entire cross-sections of complex components without any prior structural information, like CAD data, with controllable reconstruction distortion and using limited probe positions. The feasibility of this method lies in the fact that gFD-RTM combines multiple incident directions during imaging with the attitude optimization of the probe. With this strategy, the initial probe position is established as the origin of the global coordinate system from which movement in the *x* and *y* directions is used to establish a background image onto which the results of each subsequent FMC imaging process are superimposed to obtain a complete global profile imaging result. Meanwhile, a hybrid extrapolation method is applied to select subsequent probe positions and attitudes adaptively based on the current shape of the workpiece, while the automation system ensures that the probe position is sufficiently accurate to allow the reconstruction of images acquired at multiple angles. Therefore, one of the aims of this study is to provide a practical demonstration that complete and accurate cross-sectional images can be rendered from different angles of incidence for curved surface profiles. Accordingly, the proposed method is validated based on its application to an aluminum alloy test block with complex contours.

## 2. Materials and Methods

### 2.1. Reverse Time Migration Algorithm

The two-dimensional (2D) wave equation can be expressed in the frequency domain as
(1)$$\left[\nabla^{2}+\frac{\omega^{2}}{{c\left(x,z\right)}^{2}}\right]p\left(x,z,\omega \right)=s\left(x,z,\omega \right),$$
where *x* and *z* are coordinates in the horizontal and depth directions of the 2D space domain, respectively, *ω* is the angular frequency, *c*(*x*,*z*) is the sound velocity distribution, *p*(*x*,*z*,*ω*) is the sound field intensity, and *s*(*x*,*z*,*ω*) is the sound field of the source. We then discretize *c*(*x*,*z*) into *L* = *N_x_* × *N_z_* grid points, where *N_x_* and *N_z_* are the number of grids in the *x* and *z* directions, respectively. The discretized wave equation can now be rendered in the following form:(2)$$A\left(\omega \right)P\left(\omega \right)=S\left(\omega \right),$$
where **A**(*ω*) is an *L* × *L* complex impedance matrix of the imaging area, **P**(*ω*) is the *L* × 1 discrete spatial vector form of *p*(*x*,*z*,*ω*), and **S**(*ω*) is the *L* × 1 discrete spatial vector form of *s*(*x*,*z*,*ω*). The matrix **A**(*ω*) can also be regarded as the modeling operator for the forward propagation of the wave field, which can be given as **A**(*ω*) = ∇^2^ + *ω*^2^/*c*^2^.

In the forward modeling process, **S**(*ω*) is determined by the excitation signal and **A**(*ω*) is determined by the acoustic properties of the material. Then, a unique **P**(*ω*) can be obtained by solving Equation (2) using the lower–upper (LU) decomposition method. Moreover, the decomposed **L** and **U** matrices can be used to quickly solve the new **S**(*ω*), which enables wave fields that are excited multiple times in the same solution area to be effectively solved. Accordingly, this approach is ideal for the FMC process.

In the case of multiple excitations, Equation (2) can be expressed in terms of the LU decomposition as follows:(3)$$L\left(\omega \right)U\left(\omega \right)[P_{1}\left(\omega \right){,P}_{2}\left(\omega \right),\ldots ,P_{n}\left(\omega \right)]=[S_{1}\left(\omega \right),S_{2}\left(\omega \right),\ldots ,S_{n}\left(\omega \right)],$$
where *n* is the excitation source number, **L**(*ω*) and **U**(*ω*) are the LU decomposition results of **A**(*ω*), and **S***_i_*(*ω*) and **P***_i_*(*ω*) are the source and wavefield vectors corresponding to the *i*-th excitation.

Because the solution area is limited, any acoustic energy reaching the boundary of the solution area must be absorbed by adding a perfectly matched layer (PML) to the boundary, which is defined as follows:(4)$$\gamma \left(x\right)=c_{PML}cos(\frac{\pi }{2}\frac{x}{l}),$$
where *x* is the local coordinate in the PML layer, *l* represents the width of the PML layer, and *c_PML_* is a scalar.

The FMC process records the time domain signals of each excitation–receiver pair in the phased array. Therefore, a phased array probe with *n* array elements can capture *n*^2^ time domain waveforms during the FMC process. Accordingly, we define *D*(*x_r_*,*x_s_*,*t*) as the signal corresponding to the *s*-th excitation array element and the *r*-th receiving array element at time *t*. The FMC data *D*(*x_r_*,*x_s_*,*ω*) in the frequency domain can then be obtained by applying the Fourier transform to *D*(*x_r_*,*x_s_*,*t*) as follows:(5)$$D\left(x_{r},x_{s},\omega \right)=\int D(x_{r},x_{s},t)e^{i\omega t}dt.$$

It can be seen from Equation (5) that the time domain inversion signal is the complex conjugate of the original signal in the frequency domain, and the inversion process can be expressed by the complex conjugate of the FMC signal in the frequency domain as follows:(6)$$\tilde{D}\left(x_{r},x_{s},\omega \right)=D\left(x_{r},x_{s},\omega \right)*,$$
where the asterisk represents complex conjugation.

Finally, imaging is performed via cross-correlation imaging of the forward and inverse signals:(7)$$I\left(x,z\right)=\sum_{i=1}^{n}\sum_{j=1}^{N_{\omega }}\frac{R_{i}(x,z,\omega_{j})S_{i}^{*}(x,z,\omega_{j})}{S_{i}(x,z,\omega_{j})S_{i}^{*}(x,z,\omega_{j})},$$
where *R_i_*(*x*,*z*,*ω*) represents the receiver wavefields, *S_i_*(*x*,*z*,*ω*) represents the source wavefields at the *i*-th excitation, and *N_ω_* is the number of discrete frequencies in the selected frequency band. After imaging, the image is sharpened by applying the Laplace transform.

### 2.2. Probe Movement Strategy and Interface Calculation Based on a Spherical Wave Model

Application of the proposed gFD-RTM method to an arbitrary sample with a complex profile is illustrated by the schematics depicted in Figure 1 and the probe motion strategy applied under ideal conditions is described according to the flowchart provided in Figure 2. As shown in Figure 3, the first step in the gFD-RTM method is to set the sound velocity *v*_1_ of the material and the sound velocity *v*_2_ of the coupling agent (such as water or oil) so as to facilitate the establishment of a sound velocity model in subsequent work. Then, the probe is placed manually in its initial position so that the probe faces the workpiece, which serves as the global origin from which a global coordinate system is established. This initial position is also selected to allow the sound waves emitted by the probe to enter the workpiece with minimal attenuation, while ensuring that the returned sound waves have the highest signal-to-noise ratio. This initial position serves as the border upon which the termination condition of the scanning process is gauged. The second step in the gFD-RTM method is for the probe to perform an FMC acquisition and calculate the interface position based on the spontaneous and self-collecting FMC data of each array element. Then, the probe is moved to its next position if the probe has not returned to or exceeded its starting position.

The process of FMC acquisition under ideal conditions is illustrated in Figure 4a, where the FMC data are acquired as the summation of the FMC data obtained at each probe location. As shown in Figure 4b, the distance *d* between the interface and the probe can be calculated in the time domain based on the wave travel time *t* at a velocity *c*. However, this method is not applicable when *d* is calculated in the time domain for a curved surface, as illustrated in Figure 5a. As can be seen, the position of the interface wave will deviate from the axis of the array element when the reflecting interface is not perpendicular to the direction of the linear array. Therefore, only the distance between the reflector and the center point of the array element can be determined through the interface wave position in this case, as the direction cannot be determined. Hence, the interface shape cannot be accurately recovered from FMC data through time-of-flight alone.

However, the distance between each array element and the corresponding reflection point can still be calculated through the flight time by approximating the acoustic waves excited by the phased array probe as cylindrical waves, as illustrated in Figure 5b. Here, assuming that the difference between the echo time received from the interface and the excitation pulse emission time is *t*, then the distance *r* between the reflection point and the center of the array element is *r* = *t* × *v*/2, where *v* represents the speed of sound in the coupling medium. The set of possible locations of the reflection points is then a semicircle R of radius *r* with the array element as its center. A fitting method is adopted here to determine the full set of possible locations of the reflection points. The process first finds the circle R_tangent1_ that is tangent to the reflection point sets R_1_, R_2_, and R_3_ of the first three array elements. The tangent points to R_1_, R_2_, and R_3_ are p_11_, p_12_, and p_13_, respectively. Then, the circle R_tangent2_ that is tangent to R_2_, R_3_, and R_4_ of the second, third, and fourth array elements is defined, and the tangent points are denoted as p_22_, p_23_, and p_24_, respectively. By analogy, *N* − 1 circles R_tangent_ and 3(*N* − 1) tangent points *p* can be obtained. This process yields the fitting curves of all the tangent points and thereby provides accurate surface information based on a relatively simple time-of-flight method. Moreover, this process also provides the means of obtaining a relatively accurate sound velocity model for subsequent calculations.

The third step involves extrapolating the next probe position for imaging based on the current surface shape in conjunction with the basic condition that the subsequent imaging region must overlap to some extent to ensure that the entire workpiece is imaged. Curved segments are generally extrapolated based on the two main strategies illustrated in Figure 6, which include constant curvature extrapolation and tangential extrapolation. The equal curvature extrapolation method predicts the line segment based on the radius of the curvature of the known line segment and its center of curvature at point P_final_ and expands outward according to the known radius of curvature and P_final_. The tangential extrapolation method determines the tangent of the last point p_final_ in the extrapolation direction of the known line segment and expands the curve outward along the tangent. Generally, the tangential extrapolation method is most suitable for relatively flat surfaces, while the constant curvature extrapolation method is suitable for surfaces with larger curvature. Accordingly, no generally applicable method can be applied. Therefore, the present work applies a hybrid extrapolation method to select the best surface adaptively from the two curves obtained using the tangent and constant curvature extrapolation strategies based on the actual surface curvature. Accordingly, equal curvature extrapolation is used when the radius of curvature is less than 20 mm, while tangential extrapolation is used when it is greater than 20 mm.

In order to avoid a collision between the probe and the workpiece and to ensure that the probe can collect complete workpiece information after one revolution around the workpiece, different probe movement strategies are adopted on convex and concave surfaces. As shown in Figure 7a, if the center of the curvature circle of the P_final_ point is within the workpiece surface, the surface is regarded as convex. In this case, the probe first rotates to be parallel to the tangent line L_final_ of the P_final_ point and then moves 3/4 probe lengths along the tangent direction of the P_final_ point. And then, the method shown in Figure 5c,d is used again to calculate the surface shape. Pass the midpoint P_middle_ in the length direction of the probe to draw the normal vector N_middle_ in the length direction of the probe and the curved surface below the probe intersects with P_middle_. Adjust the probe angle so that it is parallel to the tangent line L_middle_ on the P_middle_ point. Finally, adjust the distance between the probe and the P_final_ point along the probe’s pointing direction (this article sets the distance to 10 mm). At this time, the adjustment is completed. As shown in Figure 7b, if the center of the P_final_ point’s curvature circle is outside the workpiece’s surface, the surface is regarded as concave. In this case, the probe first moves back 10 mm along the current probe direction and then performs the movement strategy detailed in Figure 7a.

### 2.3. Test Block Designs

Two different aluminum alloy test blocks with flat profiles were employed for experimentally validating the performance of the proposed gFD-RTM method. Flat profiles were applied to simplify the calibration of the probe. For flat surface detection, the probe needs only to be kept parallel to the plane and maintained at a constant distance from the test block. Here, the distance between the probe and the plane of the test block was maintained at 10 mm. Conducting calibration for curved surface detection requires that the probe attitude and position meet the following conditions: (1) the average distance between each array element of the probe and the interface is 10 mm; and (2) the probe must be parallel to the line connecting the starting point and end point of the interface in the imaging area. The first involved the standard test block shown in Figure 8, which was applied to verify whether the gFD-RTM method can accurately identify complex defect structures within a regular shaped object. All the side-drilled holes (SDHs) in this test block passed completely through the test block. Then, the relatively complex design illustrated in Figure 9 was tested. The test block was constructed from 6061 aluminum alloy. As can be seen, the shape of the test block was composed of multiple arcs with different diameters, and 51 side-drilled holes (SDHs) with uniform diameters of 3 mm were machined completely through the test block.

### 2.4. Experimental Setup

The experimental setup employed during testing is shown in Figure 10, which included a water tank within which the test sample was immersed and a commercial three-axis screw module. The two *x*-axes of the module are HK85-S750-F0 and are produced by Dongguan High-Tech Co., Ltd., Dongguan, China. The *y*-axis and *z*-axis are HK85-S550-F0 and HK85-S400-F0 and are produced by the same company. A vertical rotation axis module produced by Dongguan Taiji Co., Ltd., Dongguan, China installed on the *Z*-axis slider was used to control the position and attitude of the phased array probe. These translation modules were subject to ±0.05 mm, and the rotation module was subject to ±0.01°. The high-precision automated equipment ensured good correlation between the calculated probe coordinates and their physical coordinates, which is a fundamental requirement of the gFD-RTM process to guarantee accurate image registration. A CTS-PA22T ultrasound imaging system and a 5L64 probe (Shantou Ultrasonic Electronics Co., Shantou, China) were employed in the experimental setup. The 5L64 probe has 64 array elements arranged in a straight line with a 0.6 mm pitch and a 5 MHz central frequency. Both the mechanical probe positioning system and the phased array system were controlled by software operating on a host computer. The phased array system transmitted the original FMC data to the host computer for the interface calculation and imaging.

A block diagram describing the gFD-RTM process applied for imaging a test block is presented in Figure 11, where routines written in C++ were employed to move the probe around the workpiece, conduct FMC data acquisition, and perform global image matching calculations. As illustrated in Figure 11, the imaging range of a single probe position was set to an area 90 mm below the probe. As discussed above, the sequence of the probe positions must overlap to some extent to ensure that the entire workpiece is imaged. This produces a composite of overlapping images, as illustrated in Figure 12b. To this end, the initial probe position is established as the origin of the global coordinate system from which movement in the *x* and *y* directions is used to establish a background image onto which the results of each subsequent imaging process are superimposed to obtain a complete global profile imaging result. The workpiece dimensions and imaging system were selected to ensure that the imaging depth was greater than half the cross-sectional width of the workpiece, as is required to obtain full-scale composite images.

After receiving the FMC data obtained in the FMC acquisition process, the C++ routine calls another packaged MATLAB 2023a routine for the interface calculation and imaging calculation. First, the interface curve is calculated by fitting based on the spontaneous self-collection signal of each array element. Then, the imaging area is divided into two media, including the water layer and the test block material, according to the interface position from which the sound velocity model is obtained. The model is discretized, and a corresponding sound velocity is assigned to each discrete unit to generate a sound velocity matrix. The sound velocity matrix, excitation signal, and FMC signal are then applied for conducting FD-RTM computational imaging. Finally, the MATLAB routine outputs the imaging results, and the C++ routine reads the probe position and attitude from the mechanical system and matches the image to the global background image.

After completing this series of steps, a new probe position is selected and the FMC acquisition and imaging are again conducted. The defect detection process ends when the probe returns to or exceeds its starting position. At this time, the background image superimposed with all the imaging results is transformed into a complete global profile imaging result by averaging the overlapping areas of multiple images.

### 2.5. Analysis of Test Results

Because the final image obtained via the gFD-RTM method is a synthesis of multiple images, the following quantitative metrics were applied to measure the imaging quality: contrast-to-noise ratio (CNR) and array performance indicator (API). These metrics were applied according to the scheme illustrated in Figure 13, where a window centered on the defect region (green solid line) and a window in the defect-free area (red dashed line) were used for evaluating the detection performance associated with each SDH in the workpiece. Both window sizes were 5 mm × 5 mm.

Based on this framework, the CNR of each SDH was calculated using the following formula:(8)$$CNR=\frac{\mu_{signal}-\mu_{noise}}{\sqrt{{\sigma_{signal}}^{2}}+{\sigma_{noise}}^{2}},$$
where *μ_signal_* is the average pixel intensity in the window centered on the SDH, *μ_noise_* is the average pixel intensity of the window in the defect-free area, and *σ_signal_*^2^ and *σ_noise_*^2^ are the respective variances of the pixel intensities in these windows. The following formula was applied to calculate the API of each SDH:(9)$$API=\frac{A_{-6dB}}{\lambda^{2}}$$
where *A_−6dB_* is the area occupied by pixels with intensities ranging from a maximum value to 6 dB below the maximum, and *λ*^2^ is the square of the wavelength of the probe signal. The API value reflects the ability of the array acoustic transducers to detect and image point-like reflectors. Therefore, a decrease in the API corresponds with an improvement in the imaging resolution of the inspection strategy. For example, focused B-scan imaging will obtain a slightly smaller API value than planar B-scan imaging. Finally, the DPE was calculated as the difference between the expected SDH position and the position of the pixel with the maximum intensity in the window centered on the defect.

## 3. Results

### 3.1. Global FD-RTM Imaging for the Standard Test Block

The proposed scanning scheme resulted in the 14 probe positions around the standard test block illustrated in Figure 14a, and the corresponding gFD-RTM imagining results are presented in Figure 14b. As can be seen, the imaging results are complete and accurately show both arc-shaped rows of SDHs in the test block as well as other shallow and deep defects. Accordingly, the occlusion of deep structures by shallow structures has been alleviated effectively in the multi-angle imaging and superposition process, which occurs because the defect signals common to all the images are enhanced in the superposition process, while the artifact noise, which is different in each of the superimposed images, is suppressed. In addition, the capture of images from multiple angles also reduces the effects of attenuation. Finally, the amplitudes of the signals returned from each defect are similar, which simplifies the image interpretation process even without conducting distance amplitude calibration.

### 3.2. Global FD-RTM Imaging for the Complex Test Block

The proposed scanning scheme resulted in the 17 probe positions around the complex test block illustrated in Figure 15a, and the corresponding gFD-RTM imagining results are presented in Figure 15b. Naturally, the greater number of probe positions applied in this case than were applied for the standard text block was necessitated by the much greater complexity of the workpiece. Again, the imaging results are complete and accurately display the complex internal structure of the sample owing to the effective solution of the occlusion problem.

### 3.3. Analysis of the Imaging Quality Obtained for the Complex Test Block

Detailed analysis of the gFD-RTM imaging results was conducted for the individual SDHs labeled 1, 2, and 3 in Figure 15a, which are enlarged in Figure 15b, along with the defect-free region within the green box. The enlarged imaging results pertaining to SDHs 1, 2, and 3 are also presented in Figure 16a, b and c, respectively, and the corresponding one-dimensional (1D) pixel intensity distributions are presented in Figure 16d, e and f, respectively. It can be seen that the pixel intensities of the SDHs are quite large, while the intensities of the background pixels are comparatively negligible. Accordingly, we can conclude that the proposed multi-angle image superposition process obtains high contrast images.

These gFD-RTM imaging results were compared with the imaging results obtained for SDHs 1, 2, and 3 using the FD-RTM algorithm in conjunction with a local FMC acquisition process, where the probe was manually positioned for each SDH individually. The enlarged local imaging results pertaining to SDHs 1, 2, and 3 are presented in Figure 17a–c, and the corresponding 1D pixel intensity distributions are presented in Figure 17d, e and f, respectively. A comparison with the results presented in Figure 16 indicates that the local SDH images exhibit strong stripe-like artifacts not observed in the global defect images, while the maximum pixel intensities of the SDHs are at least an order of magnitude less than those obtained via the gFD-RTM method. Hence, the signal-to-noise ratio obtained under local imaging is considerably less than that obtained under global imaging.

The CNR and API metric values obtained for SDHs 1, 2, and 3 using local and global FMC imaging are presented in Figure 18a and b, respectively. These results clarify the comparisons provided by Figure 16 and Figure 17, where the CNR is generally greater and the API smaller for the global images relative to the local imaging results. At position 1, the CNR of the global composite image is decreased by 25% and the API is reduced by 67% compared with the local single imaging. At position 2, the global CNR is 53% lower than the local one, but the API is also 80% lower. At position 3, the global CNR is 71% higher than the local one, but the API is also 33% higher. Moreover, the imaging results demonstrated that an accurate imaging position, high contrast, and clear edges were obtained by the gFD-RTM method for shallow SDHs. However, the size and shape of the SDH images became increasingly distorted with an increasing depth, and the edge contrast also decreased. This is because the excitation waves used in the FMC acquisition process were cylindrical waves with no directivity. The diffusion of sound waves in the water and solid materials decreased the energy density of deep defects, such that the return signals were also significantly weakened. Here, partial defect detection is marginally successful only by superimposing the signals from several angles. The CNR and API values also demonstrate that superposition has also decreased the indication accuracy of some of the defects to some extent due to the distortion of the defect shape.

## 4. Conclusions

The present work addressed the deficiencies in current applications of ultrasonic phased-array imaging methods toward the imaging of large workpieces with complex shape features and curved surfaces by combining an automation system and phased array equipment with the gFD-RTM method. The proposed gFD-RTM method extends the capabilities of FD-RTM using multiple probe positions around the sample and adapting to its surface contours by applying a hybrid extrapolation method to select subsequent probe positions and attitudes adaptively based on the current shape of the workpiece. The proposed method was demonstrated experimentally to identify point-like and linear defects inside an aluminum alloy test block with complex contours consisting of several concave and convex surfaces. The results demonstrated that not only can the gFD-RTM algorithm render complete and accurate cross-sectional images from different angles of incidence for curved surface profiles but also improves the imaging quality over that obtained using a local FD-RTM approach.

However, the current work suffers from some limitations. For example, the magnitude and shape of the defects still vary depending on the intervening surface contours. This means that practical applications of the gFD-RTM method require careful calibration. In addition, sound velocity may be subject to subtle changes in different directions due to anisotropy and changes in the material density and internal stress. Therefore, these subtle changes must be taken into account during sound velocity model building and image processing.

## Figures and Tables

**Figure 1 sensors-24-00225-f001:**
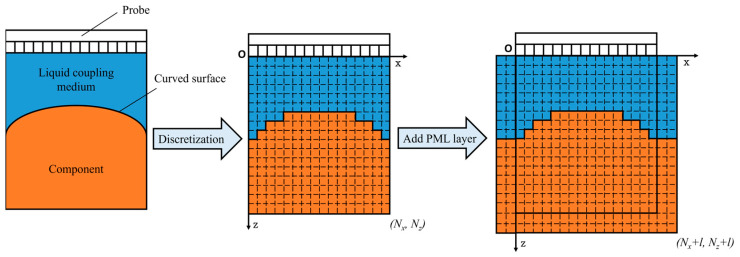
Schematic illustrating the model discretization and insertion of a PML layer.

**Figure 2 sensors-24-00225-f002:**
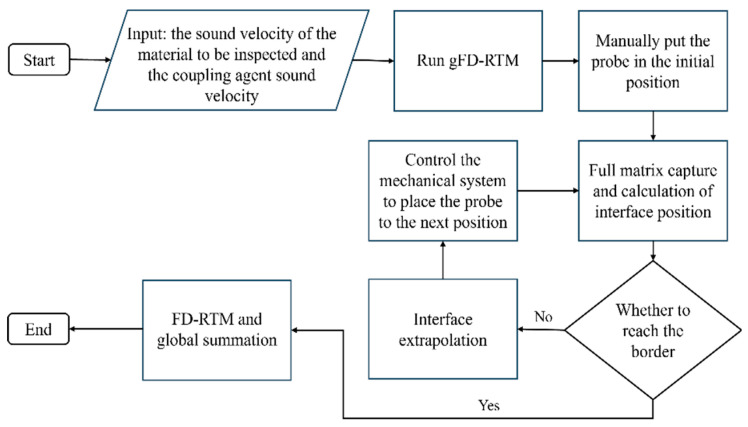
Flowchart depicting the probe positioning algorithm under ideal conditions.

**Figure 3 sensors-24-00225-f003:**
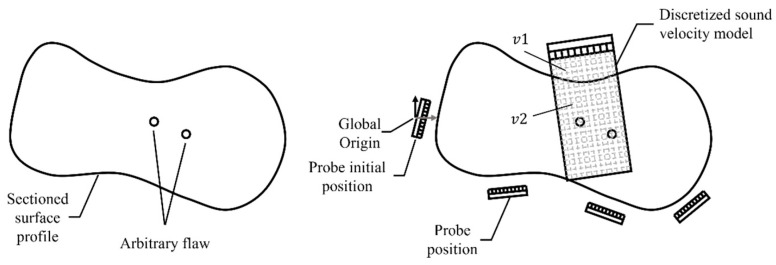
Schematics illustrating the workflow of the gFD-RTM method.

**Figure 4 sensors-24-00225-f004:**
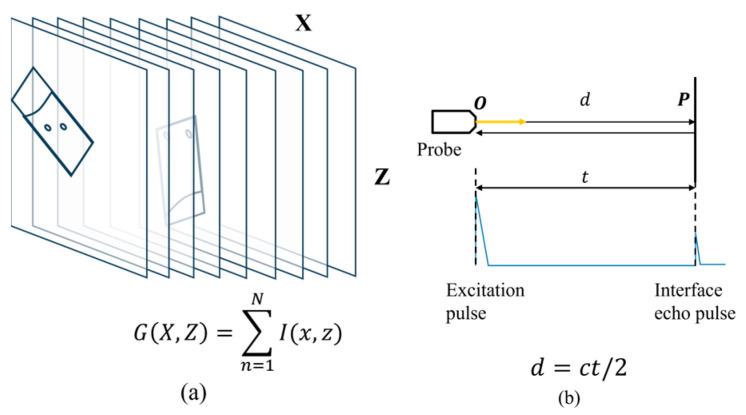
(**a**) Image summation process of the gFD-RTM method. (**b**) Interface distance *d* extrapolation in the time domain based on the wave travel time *t* at a velocity *c* under ideal conditions.

**Figure 5 sensors-24-00225-f005:**
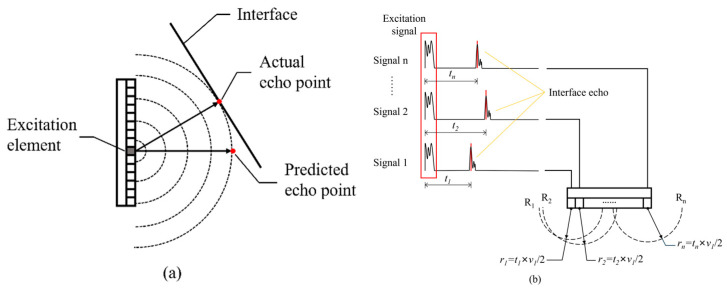
(**a**) Error in calculating the interface distance in the time domain for an inclined surface. (**b**) The reflection point set is derived from the spontaneous self-received signals of each array element. (**c**) Fitting a circle by a set of reflection points. (**d**) Calculation of the interface distance for a curved surface based on the fitting of spherical waves.

**Figure 6 sensors-24-00225-f006:**
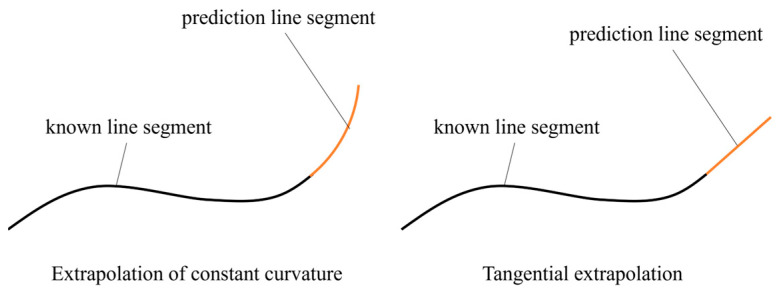
Schematics illustrating the two methods employed for extrapolating the next probe position.

**Figure 7 sensors-24-00225-f007:**
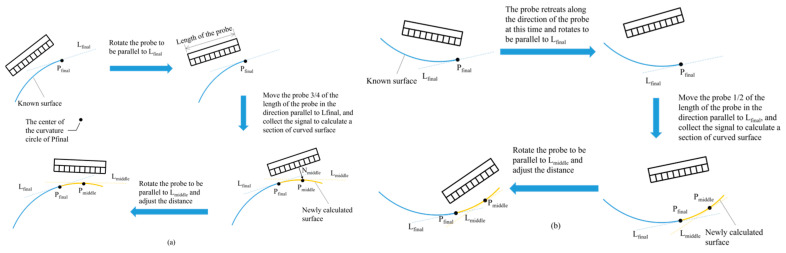
(**a**) Probe movement strategies on convex surfaces. (**b**) Probe movement strategies on concave surfaces.

**Figure 8 sensors-24-00225-f008:**
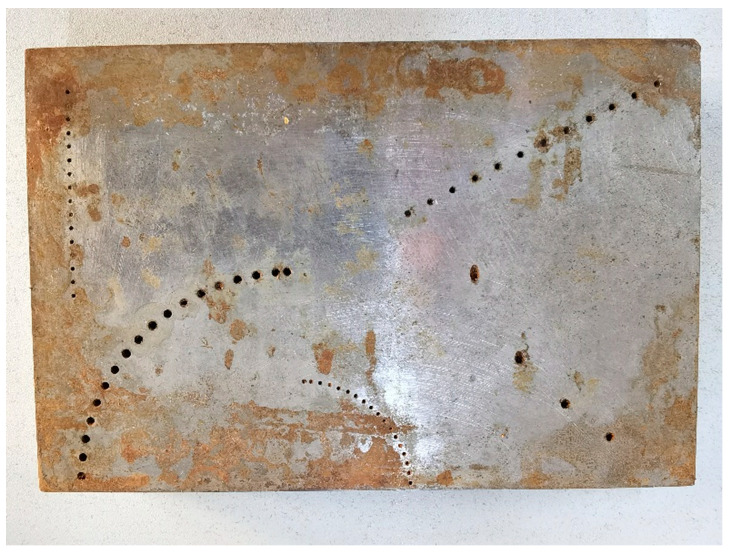
Standard test block for the phased array experiments.

**Figure 9 sensors-24-00225-f009:**
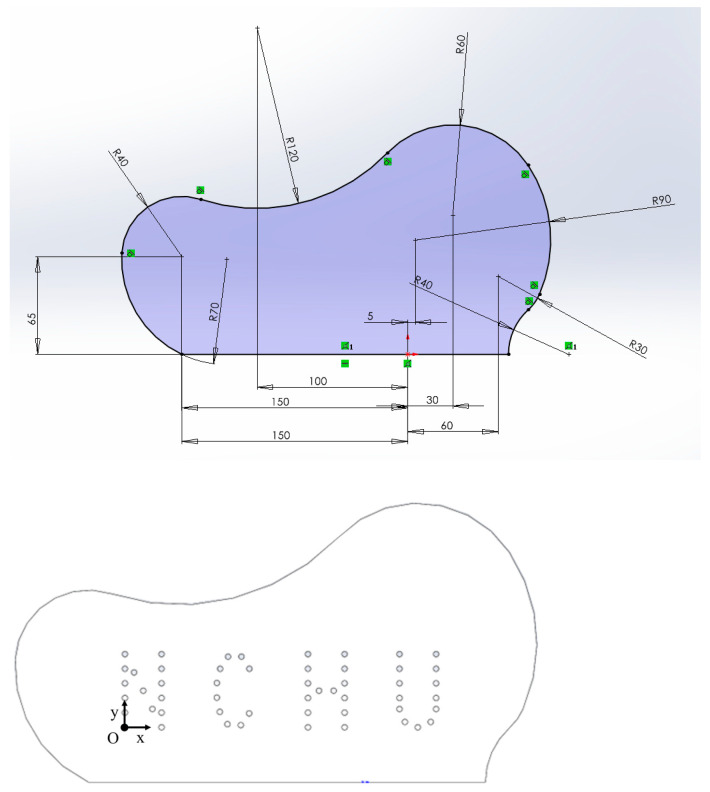
More complex test block design (unit: mm).

**Figure 10 sensors-24-00225-f010:**
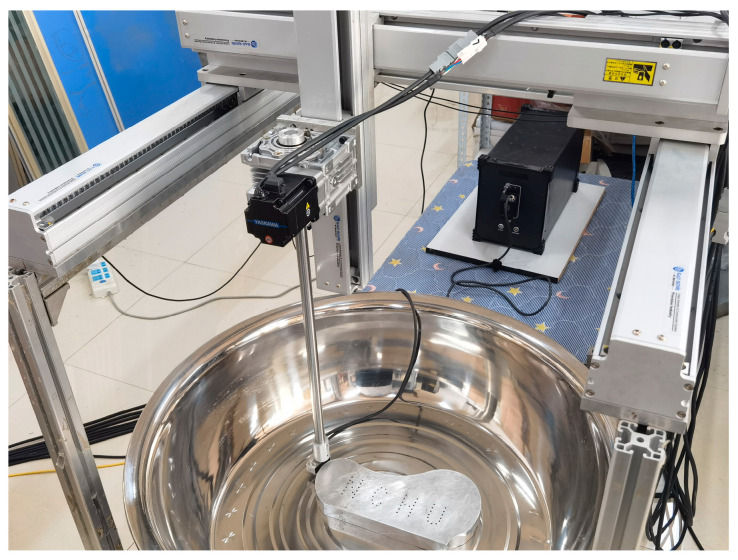
Experimental setup employed for testing.

**Figure 11 sensors-24-00225-f011:**
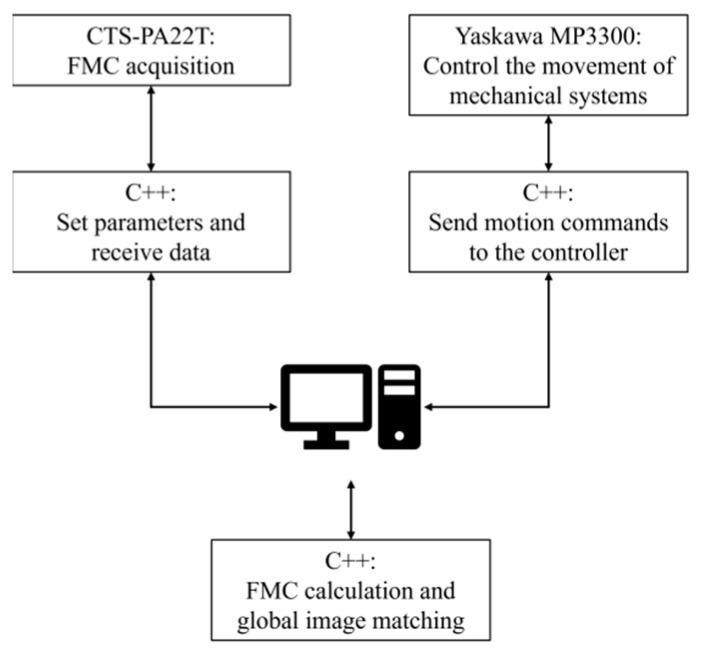
Block diagram of the gFD-RTM process applied for imaging a test block.

**Figure 12 sensors-24-00225-f012:**
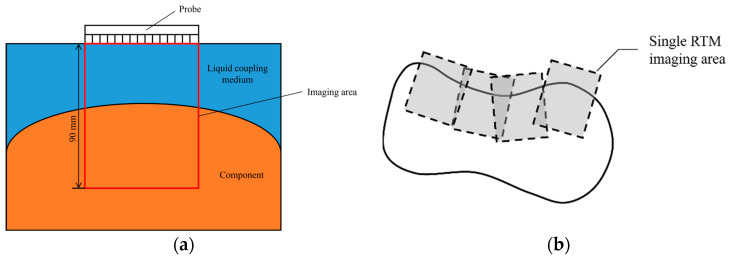
(**a**) Schematic of the imaging area. (**b**) Schematic illustrating the composite imaging area.

**Figure 13 sensors-24-00225-f013:**
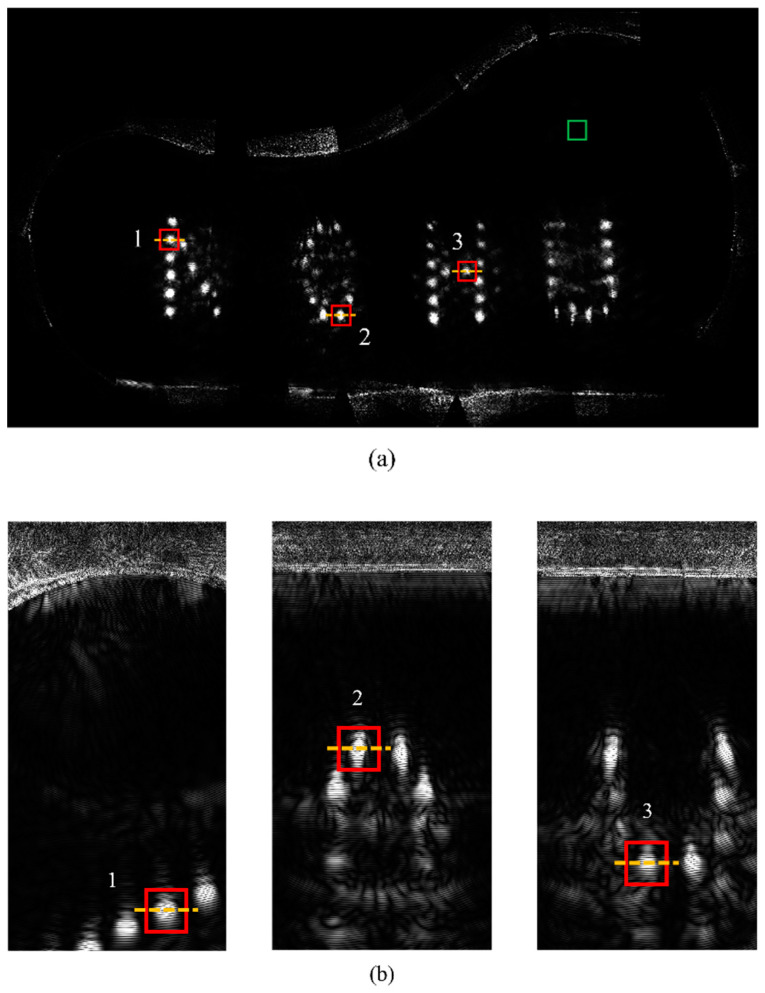
Global FD-RTM imaging results for the complex test block. The orange dashed line is the acquisition cross-section for calculating the one-dimensional pixel intensity distribution: (**a**) select SDHs within the red boxes and the defect-free region within the green box; and (**b**) local imaging of selected areas.

**Figure 14 sensors-24-00225-f014:**
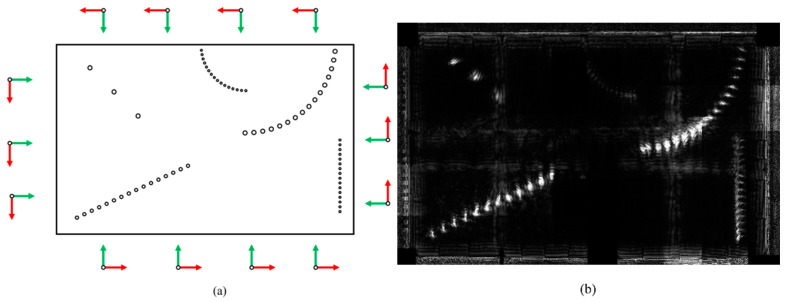
Experimental results associated with the standard test block: (**a**) phased array probe positions, the green arrow represents the direction of the probe; and (**b**) gFD-RTM imaging result.

**Figure 15 sensors-24-00225-f015:**
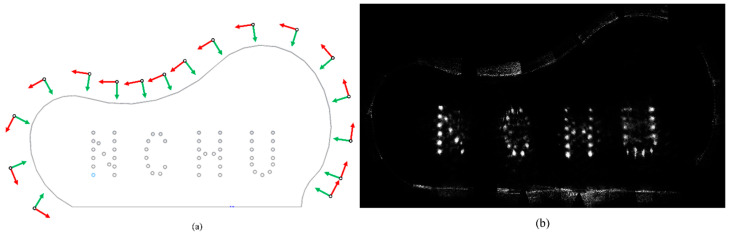
Experimental results associated with the complex test block: (**a**) phased array probe positions, the green arrow represents the direction of the probe; and (**b**) gFD-RTM imaging result.

**Figure 16 sensors-24-00225-f016:**
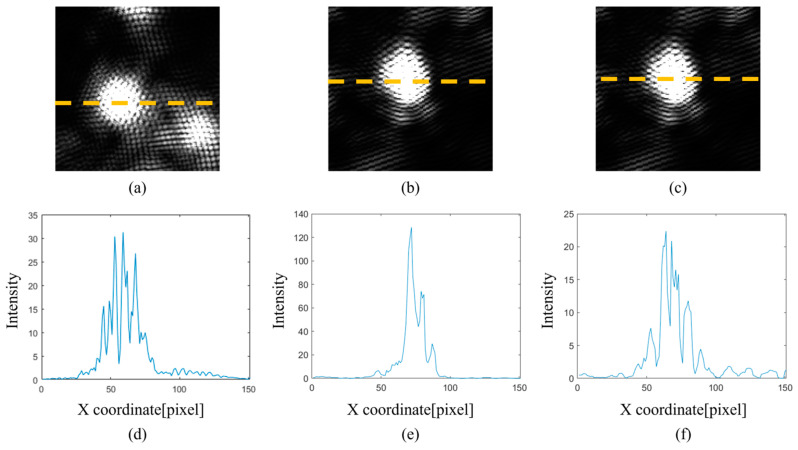
Global FD-RTM imaging and one-dimensional (1D) pixel intensity distributions of the individual SDHs presented in Figure 15b. The orange dashed line is the acquisition cross-section for calculating the one-dimensional pixel intensity distribution: (**a**) image of position 1; (**b**) image of position 2; (**c**) image of position 3; (**d**) intensity distribution of position 1; (**e**) intensity distribution of position 2; and (**f**) intensity distribution of position 3.

**Figure 17 sensors-24-00225-f017:**
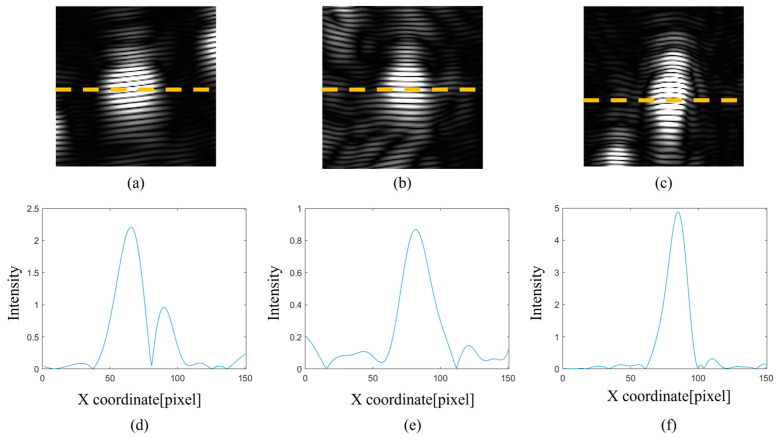
Local FD-RTM imaging and 1D pixel intensity distributions of the same SDHs presented in Figure 15b. The orange dashed line is the acquisition cross-section for calculating the one-dimensional pixel intensity distribution: (**a**) image of position 1; (**b**) image of position 2; (**c**) image of position 3; (**d**) intensity distribution of position 1; (**e**) intensity distribution of position 2; and (**f**) intensity distribution of position 3.

**Figure 18 sensors-24-00225-f018:**
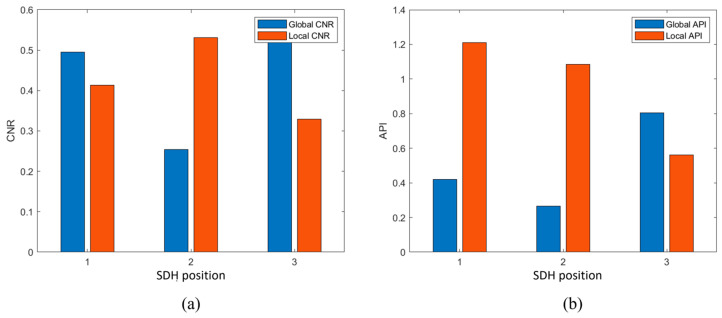
Comparison of the API and CNR metric values obtained for the individual SDHs of the complex text block presented in Figure 15b under global and local FD-RTM imaging: (**a**) API index comparison at three sampling positions; (**b**) API index comparison at three sampling positions.

## Data Availability

The data analyzed during this study are available from the corresponding author upon reasonable request with respect to ethical and privacy restrictions.

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
