# Peer review of "Adaptive Ultrasonic Full Matrix Capture Process for the Global Imaging of Complex Components with Curved Surfaces"

_sensors, 2023, doi:10.3390/s24010225_

Round 1

Reviewer 1 Report

Comments and Suggestions for Authors

The manuscript submitted by Wensong Miao et al. addresses the utilization of non-destructive testing and evaluation techniques for objects characterized by intricate shape features and curved surfaces, giving rise to complex sound beam refractions. The authors propose an innovative full matrix capture imaging method aimed at achieving comprehensive defect detection in complex workpieces. Notably, this approach does not necessitate reliance on pre-existing structural information. The methodology integrates an automated system with phased array equipment, showcasing a novel synergy that enhances global defect detection capabilities in non-destructive testing and evaluation scenarios. The manuscript is commendably well-organized, featuring a clear and concise presentation. I recommend its publication, subject to addressing the following remarks.

Remarks:

1. Refocus the abstract on the study and its results, emphasizing quantitative data over irrelevant descriptions.

2. The absence of a detailed discussion on the current state of the art in the specific topic hinders the paper's contextualization within existing literature. Consequently, it impedes the ability to make a favorable evaluation of the level of originality and innovation introduced to the field. It is strongly recommend that the authors conduct an extensive literature review to optimize the alignment of their research objectives with existing knowledge. This will enable a thorough assessment of the enhancement in measurement accuracy achieved through their proposed method in comparison to other state-of-the-art approaches.

3. In the results and discussion section, it is advisable to explicitly state the extent to which the gFD-RTM method enhances imaging quality compared to the results obtained through a local FD-RTM approach.

Author Response

Dear Reviewer

Re: manuscript reference sensors-2762060, Please find attached a revised version of our manuscript: “Adaptive ultrasonic full matrix capture process for the global imaging of complex components with curved surfaces.”

Thank you very much for your time spent reviewing the manuscript and your very encouraging comments on its merits.

Comments

“The manuscript submitted by Wensong Miao et al. addresses the utilization of non-destructive testing and evaluation techniques for objects characterized by intricate shape features and curved surfaces, giving rise to complex sound beam refractions. The authors propose an innovative full matrix capture imaging method aimed at achieving comprehensive defect detection in complex workpieces. Notably, this approach does not necessitate reliance on pre-existing structural information. The methodology integrates an automated system with phased array equipment, showcasing a novel synergy that enhances global defect detection capabilities in non-destructive testing and evaluation scenarios. The manuscript is commendably well-organized, featuring a clear and concise presentation. I recommend its publication, subject to addressing the following remarks.”

We appreciate your clear and detailed feedback and hope that we have fully addressed all of your concerns. In the remainder of this letter, we discuss each of your comments individually along with our corresponding responses.

To facilitate this discussion, we first retype your comments in italic font and then present our responses to the comments.

Comment 1:

“1. Refocus the abstract on the study and its results, emphasizing quantitative data over irrelevant descriptions.”

We agree with your suggestion and have rewritten the abstract to focus on our research and results. We have added a performance index comparison of local imaging and global imaging to the abstract.

Comment 2:

“2. The absence of a detailed discussion on the current state of the art in the specific topic hinders the paper's contextualization within existing literature. Consequently, it impedes the ability to make a favorable evaluation of the level of originality and innovation introduced to the field. It is strongly recommend that the authors conduct an extensive literature review to optimize the alignment of their research objectives with existing knowledge. This will enable a thorough assessment of the enhancement in measurement accuracy achieved through their proposed method in comparison to other state-of-the-art approaches.”

We fully agree with your suggestion and have rewritten the Introduction section to enhance our literature review. We have reduced the literature review of imaging algorithms and increased literature citations and descriptions of surface tracking technology and probe movement strategies.

Comment 3:

“3. In the results and discussion section, it is advisable to explicitly state the extent to which the gFD-RTM method enhances imaging quality compared to the results obtained through a local FD-RTM approach.”

In accordance with your suggestion, we have added a quantitative comparison of the API and CNR indicators of local imaging results and global imaging results in the results and discussion sections. The added information is as follows:

We would like to once again thank you for your time and for affording us this great opportunity to improve our manuscript. We hope you will find this revised version satisfactory.

Sincerely,

The Authors

Reviewer 2 Report

Comments and Suggestions for Authors

The article is devoted to an experimental approach based on a phased array for monitoring defects in the bulk of samples with a curved surface. The work is interesting and some moments has potential for industrial application.There are a few comments.

1. The text is difficult to read. Please improve the language.

2. Page 5 Eq(5). Probably S* should be R*, Please check.

Line 191: "where Ri(x, z, ω) and Si(x, z, ω) are the source and the receiver wavefields", Is R a receiver and S a source? Check the sequence of descriptions.

3. Table 1 does not carry any semantic load, it seems to me that it can be removed. And also Figure 11, which can be removed or combined with Figure 12.

4.  Unit 2.4. Need to add information about phase array. Geometry and dimensions, frequency range, number of elements.

5. It was not clear, how the phase array moves? What was the step of moving? Was it half the length of phased array or full length and rotation? Please comment.

6. Figure 16 and 17. Please add axes labels.

Comments on the Quality of English Language

The sentences are too long, with too many subordinate clauses. Long sentences when read sometimes cause double meaning. Some sentences are better shortened.

For example. Page 2 line 122. The idea was to show the inability of past works. Or the idea was to demonstrate the advantage of global FD-RTM in comparison with previous methods. 

Author Response

Dear Reviewer

Re: manuscript reference sensors-2762060, Please find attached a revised version of our manuscript: “Adaptive ultrasonic full matrix capture process for the global imaging of complex components with curved surfaces.

Thank you very much for your time spent reviewing the manuscript and your very encouraging comments on its merits.

Comments:

“The article is devoted to an experimental approach based on a phased array for monitoring defects in the bulk of samples with a curved surface. The work is interesting and some moments has potential for industrial application.”

We appreciate your clear and detailed feedback and hope that we have fully addressed all of your concerns. In the remainder of this letter, we discuss each of your comments individually along with our corresponding responses.

To facilitate this discussion, we first retype your comments in italic font and then present our responses to the comments.

Comments 1:

“1. The text is difficult to read. Please improve the language.”

Thank you for the detailed review. We have carefully and thoroughly proofread the manuscript to correct all the grammatical errors and typos.

Comments 2:

“2. Page 5 Eq(5). Probably S* should be R*, Please check.

Line 191: "where Ri(x, z, ω) and Si(x, z, ω) are the source and the receiver wavefields", Is R a receiver and S a source? Check the sequence of descriptions.”

Thank you for your suggestion, but after we carefully checked, the only formula for R that appeared was eq7, and we also carefully checked eq5 and eq7.

  • Eq5 is the Fourier transform formula for the time domain signal D. We believe there is no problem with this equation.
  • Eq7 is the formula for cross-correlation imaging of the transmitting wave field and the receiving wave field introduced above. We have also checked it carefully and believe there is no problem with this equation.

R in line 191 is receiver and S is source. We have replaced the phrase "where Ri(x, z, ω) and Si(x, z, ω) are the source and the receiver wavefields" with "where Ri (x, z, ω) represents the receiver wavefields, Si(x, z, ω) represents the source wavefields, and...”

Comments 3:

“3. Table 1 does not carry any semantic load, it seems to me that it can be removed. And also Figure 11, which can be removed or combined with Figure 12.”

We agree with your suggestion. Table 1 has been removed and Figure 11 has been combined with Figure 12 as follows:

Corresponding adjustments have also been made in the article.

Comments 4:

“4. Unit 2.4. Need to add information about phase array. Geometry and dimensions, frequency range, number of elements.”

We agree with your suggestion. The following information about the phase array has been added:

Comments 5:

“5. It was not clear, how the phase array moves? What was the step of moving? Was it half the length of phased array or full length and rotation? Please comment.”

In accordance with your suggestion, we have added a probe movement strategy, including the steps of moving the probe as well as the direction and distance of each step as follows:

The text description and figure were added to the end of Section 2.2.

Comments 6:

“6. Figure 16 and 17. Please add axes labels.”

In accordance with your suggestion, we have added the axes labels as follows:

Comments on the Quality of English Language:

“The sentences are too long, with too many subordinate clauses. Long sentences when read sometimes cause double meaning. Some sentences are better shortened.

For example. Page 2 line 122. The idea was to show the inability of past works. Or the idea was to demonstrate the advantage of global FD-RTM in comparison with previous methods. ”

Thank you for your suggestion on improving the readability of our manuscript. Accordingly, we have shortened some clauses that may cause ambiguity.

We have also rewritten the abstract and introduction sections to make them more relevant to the content of the article.

We would like to take this opportunity to thank you for all your time and for affording us this great opportunity to improve our manuscript. We hope you will find this revised version satisfactory.

Sincerely,

The Authors

Round 2

Reviewer 1 Report

Comments and Suggestions for Authors

The authors have well revised and enhanced the quality of the article.